# RPL Routing Protocol Performance in Smart Grid Applications Based Wireless Sensors: Experimental and Simulated Analysis

**Shimaa A. Abdel Hakeem [1,2]**, **Anar A. Hady [2,3]** and **HyungWon Kim [1,]***

1   MSIS (Mixed Signal Integrated Systems) Lab, School of Electronics Engineering, Chungbuk National University, Cheongju, Chungbuk 28644, Korea; shimaakotb@chungbuk.ac.kr
2   Electronics Research Institute (ERI), El Tahrir st, El Dokki, Giza 12622, Egypt
3   School of Engineering and Applied Science, Washington University in St. Louis, St. Louis, MO 63130, USA; anar@wustl.edu
*   Correspondence: hwkim@cbnu.ac.kr; Tel.: +82-01096606110

**Abstract:** The Advanced Metering Infrastructure (AMI) is one of the Smart Grid (SG) applications that used to upgrade the current power system by proposing a two-way communication system to connect the smart meter devices at homes with the electric control company. The design and deployment of an efficient routing protocol solution for AMI systems are considered to be a critical challenge due to the constrained resources of the smart meter nodes. IPv6 Routing Protocol for Low Power and Lossy Networks (RPL) was recently standardized by the IETF and originally designed to satisfy the routing requirements of lossy and low power networks like wireless sensors (WSN). We have two kinds of AMI applications, on one hand AMI based WSN and on the other hand AMI based PLC communication. In this paper, we proposed a real and simulated implementation of RPL behavior with proper modifications to support the AMI based WSN routing requirements. We evaluate RPL performance using 140 nodes from the wireless sensor testbed (IoT-LAB) and 1000 nodes using Cooja simulator measure RPL performance within medium and high-density networks. We adopted two routing metrics for path selection: First one is HOP Count (HC) and the second is Expected Transmission Unit (ETX) to evaluate RPL performance in terms of packet delivery ratio; network latency; control traffic overhead; and power consumption. Our results illustrate that routes with ETX calculations in low and medium network densities outperform routes using HC and the performance decreases as the network becomes dense. However, Cooja implementation results provides an average reasonable performance for AMI with high-density networks; still many RPL nodes suffering from high packet loss rates, network congestion and many retransmissions due to the selection of optimal paths with highly unreliable links.

**Keywords:** smart grid; routing protocols; objective function; RPL performance; testbed

## 1. Introduction

The Smart Grid (SG) is the application of communication and information technology to the energy grids to manage the generation, delivery and consumption of the electricity. Figure 1 shows the system structure of the SG network.

The Advanced Metering Infrastructure (AMI) is the new part of the Smart Grid which supports the transfer of two-way power and a high data rate by connecting Smart Meters (SMs) at user's homes to the Meter Data Management Systems (MDMS) in order to collect and manage data. The main structure used for the AMI network is made of one Data Collector (DC), which acts as a gateway between the gathered information from SMs at home's and the utilities companies. In this paper, we are

focusing on the communication network between data collectors and smart meters using wireless sensors environment.

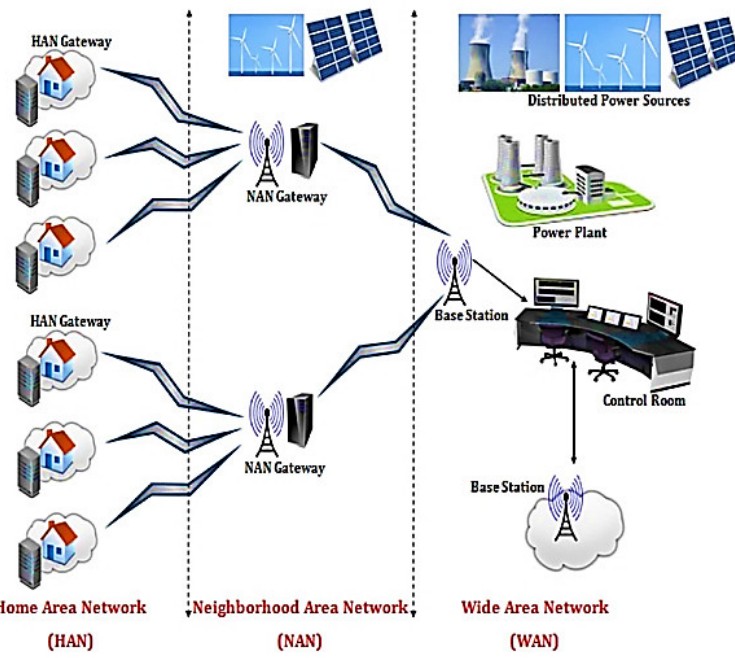

**Figure 1.** Smart Grid Communication Network.

In the PLC (IEEE P1901.2) standard [1] networks, it is impossible and very hard for most of the nodes to communicate directly due to many reasons: noises distance, etc. So, each node should collaborate by relaying different frames to have the ability to reach all nodes in the network. Many routing protocols are designed to select, maintain and construct the best paths to route packets from source to final destination. Routing process in (AMI) networks is considered as a critical issue when designing its communication network. A routing protocol survey is presented in [2] that focused only on two different communication infrastructure components of the SG, namely, Neighborhood Area Networks (NANS) and Home Area Networks (HANS). Routing protocols for HANS are categorized depending on the type of communication as some networks use Power Line Communications (PLC) technologies [3] and others use wireless communication. On the other side, routing protocols for NANS are categorized based only on the application requirements. AMI is the most challenging application of SG that utilizes NANs. The communication infrastructure design of an electric power network is considered as a hot research area which attracts both industry and academia. The large-scale communication network of SG consists of three basic parts: Access area, distribution infrastructure, and the core network. Homes, buildings and industrial collections organize the access area which is responsible for delivering, SG services to end costumers and providing user contribution to electricity production. Distribution infrastructure makes the collection of electricity usage data, and the core network is responsible for the management and control based on the received data from the aggregator center. AMI represents an example of the distribution infrastructure that connects smart meters at the user side with data aggregation center at the company side using PLC networks and wireless sensors networks (WSN) communication.

The AMI network provides information about the quality of power and quantity of consumption at the end user side. Devices in AMI networks based WSN are embedded devices with low computational and storage capability using low data rate and lossy radio communications. These kinds of networks are called Low Power and Lossy Network (LLN).

Recently, RPL protocol is considered as the most preferred IPv6 routing protocol for large-scale low power and lossy networks. The IETF proposed RPL in 2008 and in March 2012 standardization was accomplished [4]. The new release of RPL standard has been designed to support and provide the routing requirements of the AMI application based WSN. AMI is expected to support two-way communications which allow a third-party company (e.g., Electricity companies) to keep track of electricity usage, inform customers with latest prices of electricity and performs remote management within a real-time basis. The possible solution to allow all of these functionalities is to deploy a static multi-hop wireless network that connects a large number of electric smart meters to a gateway node, which in turn is directly connected to a control center which is responsible for all management kinds. AMI applications required a proper routing protocol to ensure the low-latency and the high reliability delivery for inward and outward traffic from meters to gateway and from the gateway to meters. However, smart meters' are static and fixed networks but its wireless connections still suffering from signal degradation due to the fading effects and signal interference. RPL is defined to be a Link-Layer protocol that is supposed to work within a wide of different technologies such as PLC or wireless network. RPL is part of the effort made by the Internet Engineering Task Force (IETF) to design the IPv6 architecture for low power networks (LLN), it is based on Distance Vector routing algorithms, which is designed to react and detect routing loops. RPL is originally designed for typical sensor networks such as the Advanced Metering Infrastructure (AMI) based WSN.

In order to satisfy the AMI low-latency and high-reliability requirements, the deployed routing protocol must cope with the frequent link changes by proposing effective and fast routing path selection with low routing control traffic overhead.

In this paper, we proposed experimental and simulated RPL implementations with a uniform and random network topology using IoT-LAB testbed and Java-based Cooja simulator. We proposed different scenarios with various packet sizes and different routing metrics. This study based on two hardware platform M3 Cortex nodes and Tmote Sky nodes with different experimental configuration scenarios to represent AMI network based on WSN architecture. We proposed a new RPL performance evaluation using the objective functions Hop Count and ETX, in terms of Packet Delivery Ratio, Network Latency, Control Traffic Overhead, and Power Consumption. Extensive simulations were carried out, and a detailed analysis of the proposed Objective Function ETX and OF0 to choose the best appropriate RPL configuration for the AMI applications based WSN. Adopting RPL protocol in AMI applications based PLC communication is out of this paper scope. We are interested in evaluating RPL performance and metrics to propose the most efficient objective function to choose the best path to destination.

We investigated the objective functions and the most influential parameters on RPL performance using Contiki as the wireless sensors operating system, IOT-LAB testbed, and a Java-based Cooja emulator to provide an insight into different RPL settings.

The rest of this paper is organized as follows: Section 2 provides the previous related RPL implementations for the AMI networks. In Section 3, we proposed the RPL process, RPL control messages under the consideration of AMI and RPL objective function. Section 4 presents the experimental materials and network configuration steps. In Section 5, the measured RPL metrics and the deployed scenarios are investigated. In Section 6, the results of the practical implementation and simulated implementation are conducted followed by the future work and final conclusions in Section 7.

## 2. RPL in the Literature

Routing in AMI networks is a critical challenge due to the lossy nature, constrained resources, and the routing requirements. RPL is recommended widely as the best routing protocol for lossy networks like AMI.

The evaluation and adaptations of RPL that we proposed for SG networks were based on the previous related issues introduced via evaluation and analysis of multiple RPL scenarios starting from theoretical view, simulation and experimental executions. In order to identify the problems and the existing solutions; we made an in-depth study of the state of the art to express the most recent related works found in the literature. We observed from the previous related work that RPL was always studied for WSN, no RPL studies are mentioned for the RPL behavior within PLC [5,6], and many RPL studies are based on Contiki Operating System.

### 2.1. RPL Performance Evaluation in WSN

The performance of RPL was studied in many ways as some authors build their implementations only on End-to-End (E2E) delay and Packet Delivery Ratio (PDR, while others try to measure the performance with respect to the DODAG construction issues like stability of DODAG and Trickle timer configurations.

In [7], Accettura et al. measures RPL's behavior in a WSN composed of 100 nodes. Their simulation results find that RPL network can converge quickly but they experienced a great overhead that must be reduced.

RPL's ICMPv6 control messages are studied in [8]. Generally, [8] mentioned that RPL control messages especially DODAG Advertisement message (DAO) messages may be result in network congestions. In [9], the authors investigated RPL's behavior in high density network: They find that RPL protocol guarantees a stable path, however, the physical network topology instability showed that RPL PDR is very low in dense networks. They used an indoor topology with 100 nodes on Lille SensLAB [10] testbed and they conducted experiments to prove that RPL protocol is an efficient routing protocol to find the shortest routes to destinations even in a large-scale network.

At Berkeley universities, an implementation of RPL is introduced using TinyOS [11]. In [11], authors measured the RPL performance using simulations and real data and their results observed that the control messages overhead at the beginning of RPL process increased linearly with the number of nodes participating in the RPL network and then become stable as RPL process is about to be finished. Previously, many experimental evaluations investigated RPL with small network scenarios and nowadays many researchers are about to use testbeds to test large-scale network. In [12], authors used two hardware platforms, namely WSN430 Sensor Board and the MSB430 Scatter web mote to implement RPL protocol. The results of [12] showed that RPL performed well and didn't depend on the running network topology.

Authors of [13] used the discrete event simulators Castalia/ OMNET++ to propose an evaluation of RPL protocol performance depending on various network topologies with real data. In [14], Authors using the NS2 simulator proposed a new RPL multipoint to point mechanism and RPL broadcast mechanism.

In [15], RPL performance is conducted using a Contiki operating system [16] and java based network simulator Cooja. Results showed that a few RPL nodes experienced high packet loss rate and suffered from unreliability issues while average performance was reasonable for AMI networks. In [17] authors had investigated RPL performance using a wireless sensors testbed, they conducted experiments using 100 nodes with uniform distribution through IOT-LAB testbed [18]—this study was the first phase of RPL performance evaluation. The results of [17] stated that the RPL protocol works well in medium dense networks and is considered to be the first IPv6 wireless sensor routing protocol.

### 2.2. RPL and Related Routing Protocols

A few papers focused on RPL comparison against other related protocols because RPL is a new standard IPv6 protocol designed especially for low and lossy networks and all research focuses on performance evaluation. In [18], the authors compared RPL and LOADng routing protocol as they explored that LOADng can operate in a better manner in WSN networks in terms of control message overhead, PDR, and latency.

In [19], they raise a new hybrid approach of reactive and proactive and comparing LOADng and RPL protocols. The results of [19], showed that RPL can support better performances in terms of overhead, delays and memory. In [20], the authors' shows that however LOADng protocol is originally designed for lossy and low power networks, it cannot perform well comparing to RPL in terms of E2E delay and the total time of route formation.

In [21], V. Kathuria et al. made an evaluation study of a recently developed RPL for its validation for large-scale smart meter networks. They used a realistic and detailed standard packet-level simulator called Qualnet to show that RPL outperforms an ad hoc well-known routing protocol named Ad-hoc On-demand Distance Vector protocol (AODV) especially when congestion is highly probable. The results showed a lower delay and higher Packet Delivery Ratio (PDR) for RPL compared to AODV. AODV is proven to be more vulnerable than RPL; from a scalability point of view, RPL seemed to be the best choice.

### 2.3. RPL Improvements and New Metrics

Several papers have proposed new improvements in RPL protocol to enhance the performance. In [22], Ancillotti et al. proposed new modifications to RPL based on Contiki: By changing the Contiki's behavior specially routing and neighbor table, they outperformed the original implemented Contiki's RPL performance in terms of Packet Delivery by 200%.

In [23], authors proposed a permanent probing mechanism by changing the Trickle L2 algorithm to enable a faster DODAG construction by delaying the bootstrap process till a predefined link quality is reached. They showed that the Trickle L2 mechanism can achieve better network convergence while reducing the total communication overhead.

In [24], they proposed a new mechanism to find and calculate link quality instead of considering a new neighbor with a bad link quality. They find that testing many neighbors can find the best one with the best link quality instead of sticking to only acceptable ones.

In [25], a new evaluation study of RPL performance based on simulation for large-scale outdoor scenarios was performed. RPL works well concerning the level of control overhead and network delay and introduces a repair mechanism of the corrupted radio links and tuning of the trickle timer parameters to manage the control traffic overhead with respect to data traffic. Additionally, it can also make use of the local repair mechanisms for better and quicker repairing disruption of local connection than global repair mechanisms.

In [26] RPL applicability in SG monitoring systems is investigated. Results showed that RPL makes rapid changes and adaptations to any new routing knowledge thus proving to be a good and smart choice for the SG applications. We are badly in need to study RPL behavior in realistic scenarios and different environments. Different RPL metrics were proposed in previous studies to evaluate the RPL performance and it was observed that a lot of enhancement and optimizations can be done with the RPL network.

In this paper, we introduce a new RPL performance evaluation using two different scenarios of IOT-LAB testbed with 140 nodes as an experimental scenario and a simulated scenario of Cooja simulator with randomly distributed 1000 nodes. Two routing metrics will be investigated, the Objective Function 0 (OF0) that used the hop count to calculate the routes to destination and the Expected Transmission Count (ETX) that used the expected number of packet transmission to the destination. Both metrics were evaluated in terms of four performance metrics: Control Traffic Overhead, Packet Delivery Ratio, Network Latency, and Power consumption, to propose the most efficient Objective Function that appropriate the SG applications, especially AMI. Results show that ETX outperforms OF0 in large-scale scenarios and provides better routes selection and can be considered as the default objective function of RPL protocol in AMI applications based WSN.

## 3. RPL OVERVIEW

The Routing over Low Power and Lossy network (ROLL) is an IETF working group to analyze the routing requirements of applications including industrial, home, and building automation [27]. The main objective of ROLL was to develop and design the routing solutions for IP based Low Power and Lossy Networks (LLN) that have the support of a variety of link layers. LLN is composed of constrained resources embedded devices that have limited memory, low battery power, and low processing capability. RPL is considered as 6lowpan IPv6 routing protocol proposed to choose the best path to destination with the minimum cost.

In LLNs, links are lossy and may become unstable for a short time period due to a number of reasons; for example, interference. LLNs include a wide range of link layer technologies, including Bluetooth, IEEE 802.15.4, Power Line Communication (PLC), and low power Wi-Fi. RPL is an IPv6 routing protocol for low power and lossy networks designed by the IETF (ROLL) working group as a proposed standard, its used originally to satisfy the routing requirements in WSN applications. RPL is considered as distance vector routing protocol while link state routing protocols can't satisfy the limited requirements of LLNs because it consumes a lot of power and memory to save the link states. RPL is a proactive routing protocol and starts finding the routes as soon as the RPL network is initialized. RPL forms a tree called DODAG (Destination Oriented Directed Acyclic Graph) with two types of DODAG (Grounded or Floating): Grounded DODAG in which nodes send their traffic to the gateway node and the gateway will forward them to the destination on behalf of them, while Floating DODAG has no gateway node and each node is responsible to forward its traffic. Each node has a rank value that is calculated with respect to the gateway node according to a predefined cost metric like hop count, bandwidth, reliability or number of transmissions. Each node in an RPL network has a preferred parent which works as the gateway of this node to the destination. If an RPL node didn't find any path in its routing table for a packet, the node forwards the packet to its preferred parent and so on until it either reaches the destination or a common parent which forwards it down the tree towards the destination. Nodes in an RPL network must have routes for all the nodes down the tree. It means that the nodes which are near to the root node must have large routing table entries. Route aggregation is not recommended because of several problems in LLN like the mobility of nodes and the losses due to the radio medium.

### 3.1. RPL Topology and Control Messages

As shown in Figure 2, an RPL DODAG is a DAG graph rooted and pointed at a single destination while the DODAG root has no edges. A DODAG graph is identified uniquely using a combination of DODAG ID and RPL Instance ID as the RPL instance can contain more than one DODAG each one of them has an DODAG ID. Each node in the DODAG has a rank value which expresses the node position with respect to the DODAG root node. Rank values strictly increase towards down direction and decrease in the up direction as it becomes closer to the root node. DODAG Root is responsible for the routes aggregation and DODAG construction. Traffic towards the root node is considered as Multi_point to Point (MP2P) while the traffic from root node to leaves nodes considered as Point to Multi_point (P2MP). In our proposed scenarios, we consider RPL topology with one DODAG root and one instance.

RPL protocol has three types of ICMPv6 control messages, which are defined as follows:

- DODAG Information Object (DIO): The DODAG root (border router node) issues DIO message in a multicast form to construct a new DODAG. The DIO message structure contains all information concerning the network that allows any node to find an RPL instance, select a DODAG parent set, learn about its configuration parameters, and finally build the DODAG.

- Destination Advertisement Object (DAO): As the DODAG is being constructed each node in the DODAG sends this message to propagate and populate a node rank and routing tables' information to their predecessor nodes that support the downward route traffic (traffic towards leaves nodes).
- DODAG Information Solicitation (DIS): These messages are sent by any node to trigger the others to send DIO messages to this node and this happened only when that node didn't receive a correct DIO message for a long time.

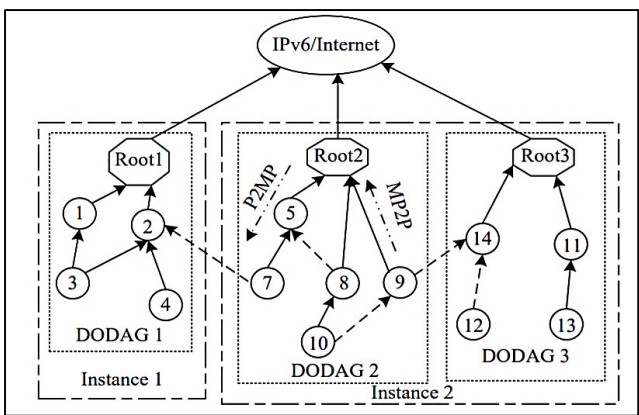

**Figure 2.** RPL DODAG Construction.

*3.2. RPL Topology*

The root node starts building a new RPL instance by issuing a (DIO) message. A DIO message provides all information about the DODAG, which can be summarized as follows:

- A DODAGID: Used to identify the root node and it's corresponding DODAG.
- Nodes use the rank value to calculate their positions in the DODAG with respect to the root node and other nodes.
- Objective Code Point (OCP): Used to identify the Objective Function needed to calculate the DODAG rank depending on the predefined constraints and metrics, which is used within the DODAG construction.
- After receiving the first DIO message, the receiving node adds the address of its sender to its parent list and computes the rank value of this sender according to the proposed objective function.
- Keep in mind that the node's rank must be greater than its parents rank as shown in Figure 2.
- Each node chooses the most preferred parent and forwards the DIO message in a multicast manner with the new rank information to the other nodes. When an RPL node receives a DIO message, it can do one of three things:

  1. Discard this message with respect to additional criteria set by RPL mechanism.
  2. Analyze the received DIO message to decide if it will maintain its existing location or change it to another lower depending on the path cost and the Objective Function proposed within the DODAG.
  3. To avoid occurrence of routing loops, a node must drop all nodes in its parent's list with ranks lower than the new calculated node's rank. After the DODAG has been constructed, each node would have a defined route formed by the most preferred parents to the destination root node, Figure 3 summarizes all steps after receiving a DIO message.

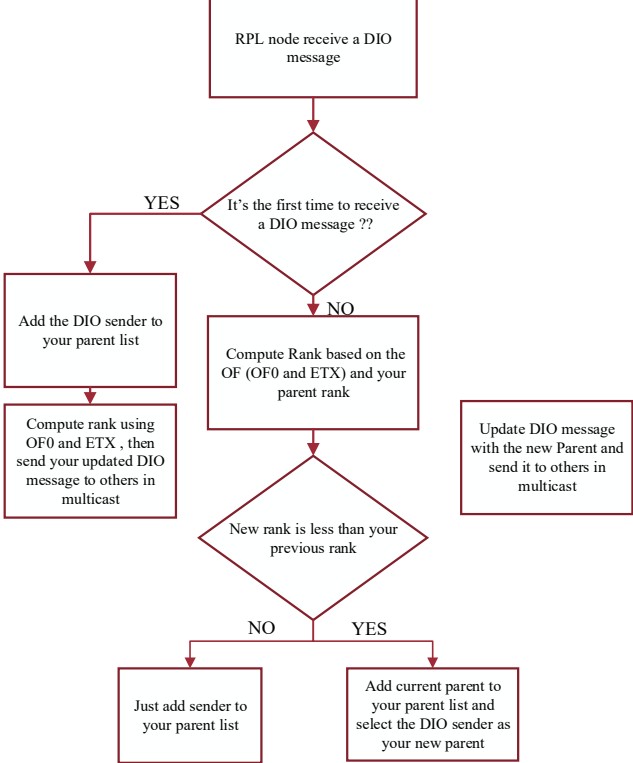

**Figure 3.** RPL DIO Message Mechanism.

### 3.3. RPL Objective Function

The Objective Function is used to define one or more metrics to help RPL nodes translate these metrics into ranks. It is responsible for selecting and optimizing the routes in a DODAG. Rank computation is carried out using the Objective Function depending on defined routing metrics such as link quality, delay, and connectivity. The Objective Function is used to define the rank of a node, which is expressed as the node distance from a DODAG root node. RPL main specification has no default objective function. Therefore, Objective Function 0 (OF0) is considered the default function, which is common to all implementations. The efficient objective function's design is still a critical research issue. In this paper, we are focusing on the evaluation of two objective function's implementations: One using Hop Count (HC) and the other using Expected Transmission Count (ETX) as a link metric in the rank calculation.

Hop Count (HC): It is the metric used to identify the number of hops from source to destination. In Contiki operating system, Objective Function zero (OF0) is the default OF and selects the path to the root with the minimum number of hops. Contiki uses a 16-bit rank value in units of 256 that allows a maximum of 255 hops. Each node calculates its rank with respect to its parent rank using the summation of the parent rank and the (default-min-hop-rank-increase) value that is defined as 256 in the RFC (6550) [28]. The rank calculation based on hop count Objective Function can be calculated as follows:

$$R(n) = R(P) + (\text{default\_min\_hop\_rank\_increase}). \tag{1}$$

while R(n) represents the rank of the node n and R(P) is the rank of the node parent, node (n) selects the parent node that minimizes the value of R(n)

Expected Transmission Count (ETX): ETX is defined as the expected number of transmissions which are required to send a packet over the communication link. The path ETX is the sum of the ETX of all the links along the path. When ETX is applied, the nodes must select the parent that has the lowest ETX value. Each node uses ETX to calculate the path to the root node and select its parent, which has the minimum overall ETX to the root node. ETX over a link can be calculated as follows:

$$\mathrm{ETX} = 1/(\mathrm{DF} \cdot \mathrm{DR}). \tag{2}$$

DF represents the probability of receiving a packet from the neighbor node and DR is the probability of receiving an acknowledgment successfully. Node (n) can calculate its rank based on ETX using this formula:

$$\mathrm{R(n)} = \mathrm{ETX} + \mathrm{R(P)} \tag{3}$$

where R(P) represents the rank of the parent node, so the total node rank can be expressed as the rank of its parent with the addition to ETX of the overall path.

## 4. Experimental Materials

In this paper, we propose two different scenarios based on actual real testbed (IoT-LAB) and Cooja simulator to expand the RPL evaluation environment and platforms. The following sections will explain each one of them.

### 4.1. Description of IoT-LAB testbed

We used a set of 140 M3 nodes from IOT-LAB wireless sensor testbed to conduct experiments by introducing a static multi-hop AMI network, which consists of n smart meter nodes and one gateway node. The root node was chosen to be in the middle of the network, as the nodes positions affect the total time consumed by all nodes to join the RPL DODAG. During experiments, the Contiki operating system is used to program the wireless sensor nodes as it's designed for low power and lossy networks. Monitoring and controlling nodes are done using Foren6 [29] that is considered as the network monitoring tool developed specially for IOT-LAB testbed. IoT-LAB is a very large-scale testbed suitable for experiment and tests tiny wireless sensor nodes and communicating objects. IoT-LAB infrastructure is located at six sites in France and allows controlling access to 2728 sensor nodes. In this paper, a subset of 140 nodes of Lille-Nord Europe site was used. Lille-Nord Europe is consisting of 332 Cortex-M3 open nodes uniformly deployed over a 225 m2 area as nodes dispatched on wooden poles and ceiling. Lille testbed is deployed over the three floors of Inria building, through corridors, offices, storage or meeting rooms. In Figure 4 Lille physical topology over the ceiling and wooden pools are expressed as nodes deployed on the ceiling over a 1.20 m × 1.20 m grid, at 9.6 m high and hanged vertically on the poles at an overall height of 7.60, 8.50 and 9.40 m. In IoT-LAB testbed there are various node types like an open node, a gateway node, and control node. Both control and gateway are used to monitor and control the open node by monitoring the power consumption and select the power supply during experiments. While the open node is totally open and fully accessed by the user (e.g., any operating system can be loaded, compiled and debugged). The gateway node can handle, control and gather open node sensors' information through a serial link. In this study, a new generation of open nodes called M3 is used with the addition of a combination of the control node and gateway node on the same ship called the host node. Table 1 summarizes components of M3 open node and host node for more details consults M3 cortex node datasheet [30].

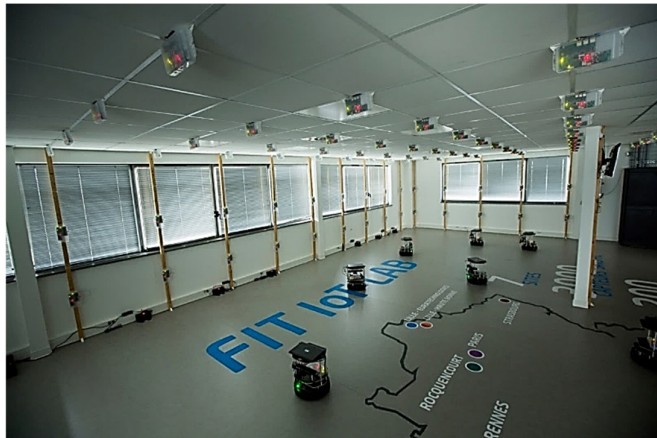

**Figure 4.** Lille–testbed network grid over ceiling and wooden poles in Inria building floors.

**Table 1.** IoT-LAB wireless open and host node components.

| Node Type | Component | Description |
|-----------|-----------|-------------|
| Open Node | Microcontroller | ARM Cortex-M, 32 bits, 72 MHz, 64 KB RAM. |
|  | Radio chip | 2.4 GHz, transmitted Power +3 dBm, data rate = 250 kbps |
|  | Power | (LiPo battery), 3.7V, 650 mAh |
| Host Node | The main- processor | VAR-SOM-AM35 CPU. |
|  | Radio chip | 2.4 GHz, transmitted Power +3 dBm data rate = 250 kbps |
|  | Power | Powered over Ethernet (PoE) |
|  | The co-processor | ARM Cortex M3, 32 bits, 72 MHz, 64 KB RAM. |
|  | Connectivity | USB ports, Ethernet ports |

The IoT-LAB testbed experiment steps are summarized as follows:

- Flash up sensors nodes with the Contiki binary images and control nodes using their serial lines. In this study, three types of firmware have been compiled to flash the nodes up (e.g., Root node, Collector node, and Client); all of them are compiled with the appropriate parameters and saved to binary images which can be uploaded to the nodes.
- The position of nodes is very important as the border router should be in the middle of the experiment area and the collector nodes (sniffing nodes) must be positioned to have the ability to sniff and monitor all nodes traffic.
- Sensor nodes are accessed directly using web access over IPv6 from the local machine by using a local Contiki Tunslip6.
- Tunneling process is very important to access the IPv6 network of IoT-LAB testbed within internet and other IPv4 networks
- Tunslip6 is used to bridge IPv6 network into IPv4 network and vice versa by creating a virtual interface (TUN) and uses a Serial Line Internet Protocol (SLIP) to encapsulate and pass the IP traffic to the other side of the serial line. Figure 5 shows the IoT-LAB testbed experiments steps.
- This testbed study is limited to use fixed nodes with only one border router and one RPL instance.
- IOT-LAB testbed allows users to control and monitor the running experiments through serial lines of nodes with the addition to diagnostic tool Foren6.
- Foren6 is a 6LowPan network analysis tool provided by the Belgian research center CETIC to capture RPL traffic and render the network state in a graphical user interface.

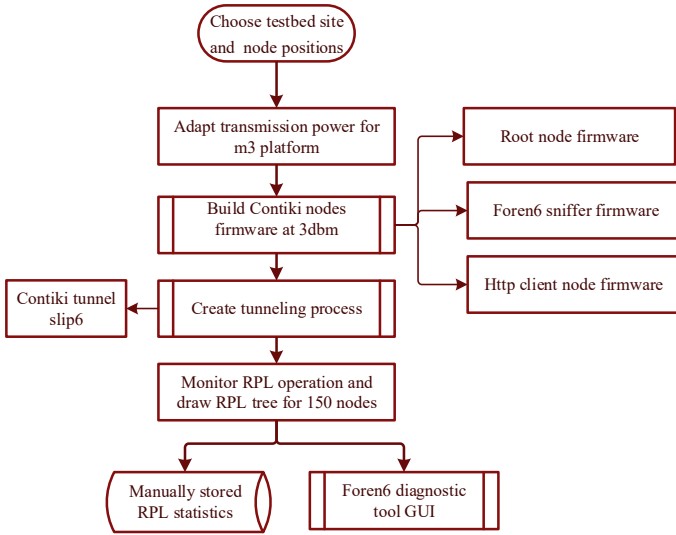

**Figure 5.** IoT-LAB testbed RPL experiment steps.

## 4.2. Description of Operating System Contiki

Contiki is an open-source operating system designed for wireless sensors networks that can handle multitasking operations. Contiki is implemented using C language. It supports Instant Contiki, which provides the necessary software tools and compilers within a Linux environment. It consists of three network stacks:

1. Rime stack: A set of lightweight networking protocols.
2. μIP TCP/IP stack: This stack provides IPv4 networking protocols.
3. μIPv6 stack: This stack offers a lightweight IPv6 protocol for tiny and embedded sensor devices. In this paper, the μIPv6 stack will be used as it consists of IPv6 protocols and RPL routing protocol and offers a Contiki MAC layer that packages radio packets into IEEE 802.15.4 frames.

## 4.3. Description of Cooja Simulator

COOJA is a Java-based simulator that is used to simulate PRL protocol, however, the nodes are programmed in C [31]. Cooja supports Contiki operating system images that are compiled natively on real sensors hardware or which are emulated using MSP430 emulator in Cooja. Cooja provides external plugins to interact with the simulated wireless nodes via Timeline, simulation Visualizer and radio logger plugins. Cooja can run real data files from testbeds and can simulate independent networks based on some defined parameters.

## 5. Performance Evaluation

In this paper, we conducted real and simulated experiments using IoT-LAB testbed, 6LoWPAN Foren6 troubleshooting tool, and Java-based Cooja simulator. In this study, we investigated different platforms with uniform and random network topology and different network size. Both scenarios are introduced in the next sections.

## 5.1. IoT-LAB Testbed Scenario

Experiments during IoT-LAB are repeated 70 times with two topologies of 140 uniform and randomly distributed M3-cortex nodes for two hours with more than 70000 IPv6 packets exchanged between nodes per one experiment run. After all, nodes are updated with the firmware, each node tries to join the RPL network and the DODAG root broadcasting DIO messages to announce the DODAG ID, root rank and the RPL configuration parameters. Each RPL node receives a DIO message and calculates its rank value with respect to the root node then adds the sender in its parent list.

RPL nodes use two objective functions, OF0 and ETX, to calculate the rank value and choose the best path to a destination. RPL tree is constructed when all nodes join the DODAG and maintain their routing tables. Figure 6 shows the logical RPL tree after all nodes joined the DODAG using Foren6 wireless sensors diagnostic tool. RPL DODAG logical topology using 6LoWPAN Foren6 diagnostic tool as the gray lines represent the radio and each circle represents the node, which contains two numbers: One of them the DODAG ID and the second is the last number of its IPv6. All details about RPL parameter configuration are presented by authors of [32,33] and Table 2 summarizes our network setup parameters.

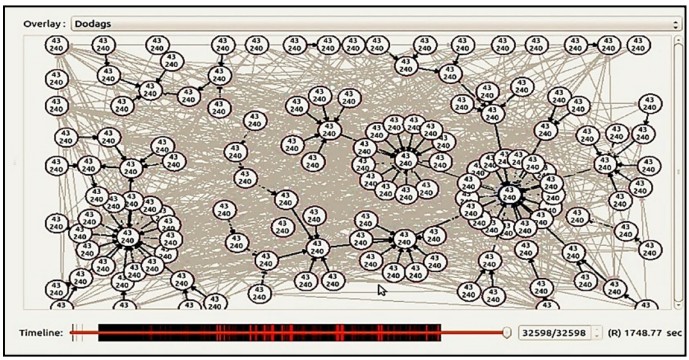

**Figure 6.** RPL DODAG logical topology using 6LoWPAN Foren6 diagnostic tool.

**Table 2.** IoT-LAB testbed experiment parameters.

| IoT-LAB Experiment Parameters | Value |
| --- | --- |
| Num. of nodes | 140 nodes M3 hardware |
| Network Topologies | uniform and random over 225 m$^2$ |
| Testbed site | Inria Lille |
| Experiment duration | 120 min |
| Operating system | Contiki 2.7 |
| Frequency | 2.4 GHz |
| Mac protocol | Contiki- MAC protocol |
| Transmission power | 3 dBm |
| Startup delay | 65 s |
| Data rate | 250 kbps |
| The number of collectors | 7 sniffers |
| The number of the border router | one node |
| DIO Interval Minimum(ms) | 12 (controls the rate of DIO message transmission) |
| DIO Interval Doublings(ms) | 8 (the number of times, the DIO minimum value can be doubled) |
| RPL Mode Operation | 2 (storing mode) |
| DIO Redundancy Constant | 10 |
| Max Rank Increase | 1792 |
| Minimum Hop Rank Increase | 256 (the root rank is 256) |
| DATA packet period | 45 s |

## 5.2. Cooja Simulated Scenario

The second investigation is based on a simulation environment using Cooja simulator to measure the RPL protocol performance in large-scale networks. A large-scale scenario of SG network is simulated using 1000 motes to emulate smart meters that are a small constrained device to report power usage to a central authority. A smart meter is essentially a sensor node in charge of collecting data and finally data routing. Cooja simulator is used to emulate Tmote Sky platform with MSP430 board with a compatible CC2420 radio chip. The Multipath Ray Tracer Medium (MRM) model is considered as the link failure model, which is used to make a realistic interference. This link failure model presents the ray tracing techniques to model different types of radio propagation effects, such as diffraction, multipath, and refraction. Nodes are dispersed randomly and uniformly over an area of $300 \times 300$ m$^2$ and each node sends a data packet of 200-bytes every 30 s towards the root

node. We repeat simulation for different values of RX ranging from (30, 40, 50–100%). Traffic logs are extracted from the simulation project using a Perl script. Table 3 summarizes Cooja experiment parameters. The objective of this experiment is to evaluate OF0 and ETX for only the upward traffic defined from client nodes to the root node. We run simulations for different RX values ranging from (30, 40, 50–100%).

**Table 3.** Cooja simulator RPL configuration Parameters.

| Parameters | Value |
| --- | --- |
| Num. of nodes | 1000 nodes |
| Network Topologies | Randomly distributed |
| Experiment duration (S) | 1300 |
| Operating system | Contiki |
| Frequency | IEEE 802.15.4 radio |
| Mac protocol | Contiki MAC |
| Startup delay (s) | 65 |
| Data rate transfer | 200-byte payload every 30 s |
| The number of the border router | One border router |
| DIO Interval Minimum(ms) | 12 |
| DIO Interval Doublings(ms) | 8 |
| RPL Mode Of Operation | storing mode |
| Use Authentication | No |
| Path Control Size | 0 |
| DIO Redundancy Constant | 10 |
| Max Rank Increase | 1792 |
| Minimum Hop Rank Increase | 256 |
| OF | OF0 & ETX |

*5.3. The RPL Performance Metrics*

In this section, we introduce four main metrics to measure RPL performance while adopting two different objective functions (OF0, ETX). These metrics provide an important impact on the routing process as RPL performance is changing according to the network topology and the hardware platform. We measured the following RPL metrics:

- **Control Traffic Overhead**: RPL has three different control messages as described before (DIO, DAO, DIS), the total of all control messages sent from node clients to the root node measures the control traffic overhead metric.
- **Packet Delivery Ratio (PDR)**: This metric is used to measure the total received packets at the root node compared with the total sent packets from clients. The higher PDR percentage of successfully delivered messages to the root node during the test, the higher routing protocol performance.
- **Network Latency (End-to-End delay)**: It is considered as the total time taken for a packet to travel from client node to the root node (destination).
- **Energy Consumption**: Power estimation is a very important factor in lossy networks as it indicates the lifetime of WSN. In this study, we use a percentage radio on time of the radio to reflect the consuming power in the sensor nodes.

## 6. Results and Evaluation

Results of IoT-LAB testbed and Cooja simulator will be shown in the following sections.

*6.1. IoT-LAB Realistic Scenario*

In this study, using IoT-LAB realistic testbed we provide two uniform and random network topologies to study RPL objective function. OF0 and ETX are investigated with respect to Control Traffic Overhead, Packet Delivery Ratio, Network Latency, and Power Consumption. We measured the RPL performance metrics simultaneously to select the best routes and optimize the network packet

delivery ratio without consuming more energy. The results of both uniform and random IoT-LAB topology are expressed in the following section.

■ Control Traffic Overhead

Figure 7 illustrates the RPL Control Traffic messages (DIO, DIS, DAO) with uniform network topology using OF0 and ETX. In general, the low-density networks provide fewer control messages than dense networks. When RPL protocol uses OF0 to calculate the route to the destination, it provides too many control messages rather than using ETX. This scenario is considered as a medium scale network that experienced a high control overhead to build the DODAG routes, which is used to propagate the routing tables between nodes. We note that DIO messages are the dominant control messages between all RPL messages as they are used to update and construct the routing tables. The increasing of control messages reflects the network instability and the increasing of the collision and congestion between packets. This collision increases the network delay and consumes the network resources. The RPL scenario, which uses OF0 to calculate the routes, consumes more control traffic overhead than scenario using ETX.

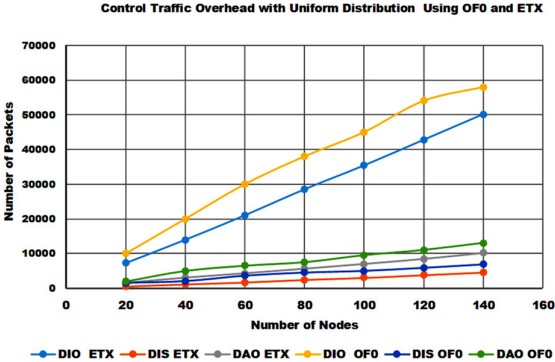

**Figure 7.** IoT-LAB testbed control traffic overhead with uniform network topology.

Figure 8 illustrates the RPL Control Traffic messages (DIO, DIS, DAO) with a random network topology using OF0 and ETX that indicates higher control messages than uniform topology.

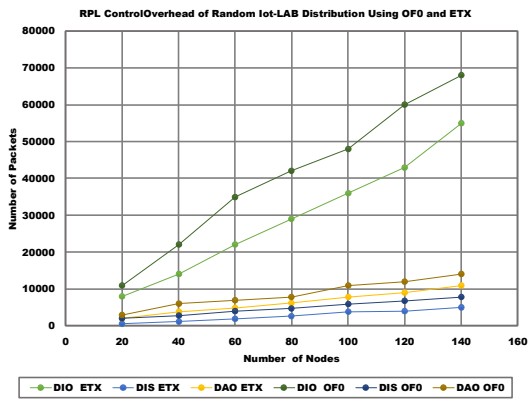

**Figure 8.** IoT-LAB testbed control traffic overhead.

Figure 9 shows the comparison between random and uniform topology with OF0 and ETX of total control traffic for 140 nodes. ETX outperforms OF0 in route calculation under both uniform and random topology while random networks need higher control messages due to collisions and retransmissions. Around 90000 packets are exchanged between 140 nodes for two hours experiments using OF0 and under random nodes distribution. In small densities, the control overhead metric is similar in both uniform and random distribution.

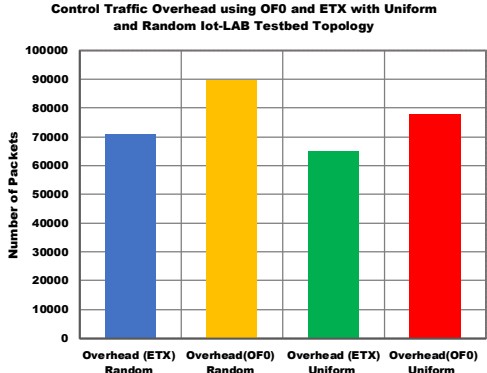

**Figure 9.** Comparison between total control traffic overhead in IoT-LAB testbed using OF0 and ETX with random and uniform network topology.

■   Packet Delivery Ratio (PDR)

We calculate the average packet delivery ratio with a uniform and random testbed topology using OF0 and ETX. PDR value is decreased while increasing the number of nodes and PDR metric for ETX is slightly higher than OF0 which reflects the ETX higher efficiency.

Figure 10 shows the PDR comparison between uniform and random testbed topology under OF0 and ETX as using ETX with a uniform topology outperforms the random topology because the random topologies always have many collisions and packet congestion that result in more retransmission. However, for a high network density, a lot of packet loss happened due to interference as nodes send many messages to some multiple destinations with the same minimum rank. From this result, ETX cannot be considered as the best solution for the high-density networks with poor link quality.

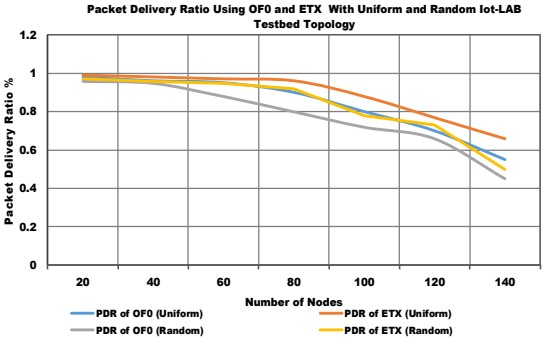

**Figure 10.** Comparison between packet delivery ratio in IoT-LAB testbed using OF0 and ETX with random and uniform network topology.

■   Network Latency (End-to-End Delay)

Figure 11 shows the comparison between the latency of ETX and OF0 with IoT-LAB uniform and random scenarios. It shows that OF0 has higher network latency than ETX because ETX considers the link details in computing the best path to destination while OF0 depends on hop count. The comparison shows that latency using ETX in uniform and random scenarios outperforms OF0 as random scenarios suffer from higher network latency while the network size increase. The average latency is calculated here after extracting the actual time of sending at client's nodes and the receiving time at root node using Foren6 diagnostic tool and command line tools of the serial lines of the testbed.

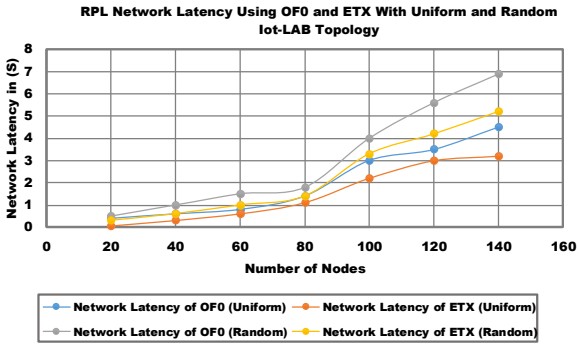

**Figure 11.** Latency comparison using OF0 and ETX with uniform and random IoT-LAB testbed.

■ Energy Consumption

Power consumption is considered as the most critical constraint of any wireless sensors network. In this study, we measure the power consumption while the RPL routing protocol is running. In this study, we develop a power trace application with the help of the energest power profile in the Contiki operating system to measure the power consumption of transmitter, receiver, idle-lessening mode, and CPU power. It's recommended measuring only the radio transceiver power as its considered as the major power consumption source in any node. Nearly the majority of power is consumed on retransmission and idle listening. In this study, we suggest the radio time to indicate and reflect the power consumption instead of considering the absolute power consumption in joules. Radio on time is the percentage of the radio during the total time of the experiment while the nodes are on for sending or receiving.

We change the network density from 20 nodes to 140 nodes to measure the consumed power, as the network becomes denser as more consumed energy. This increase of power consumption due to the more transmissions sent by the node. As shown in Figure 12, the total absolute power consumption is measured over time and the total power consumption is mainly the transceiver power while the other sources of power are very small. So, in our study we neglect the low power mode and CPU consumption. We focus on computing the % radio on time which represents the energy consumption. The more network traffic, the more consumption of the energy and vice-versa. If we send more application messages, the more energy is consumed. In Figure 13, the average power consumption is indicated for OF0 and ETX with uniform and random topologies. ETX consume less power than OF0 while random topologies need more power for both OF0 and ETX to retransmit packets. In Figure 14, the power consumption is expressed using the on radio time percentage which decreased as the packet reception ratio increased. ETX on radio time is less than OF0 in both uniform and random scenarios.

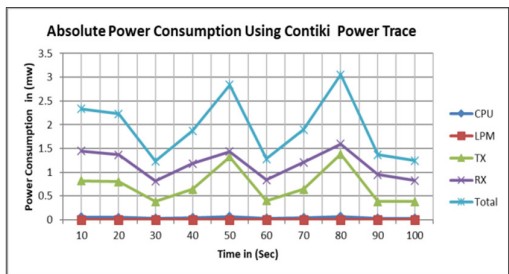

**Figure 12.** The total absolute power consumption for one testbed node measured over time.

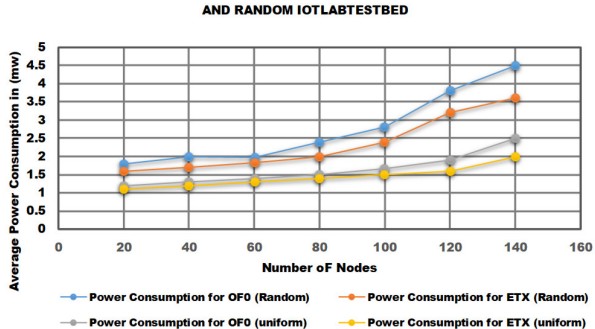

**Figure 13.** The average power consumption for OF0 and ETX with uniform and random topologies.

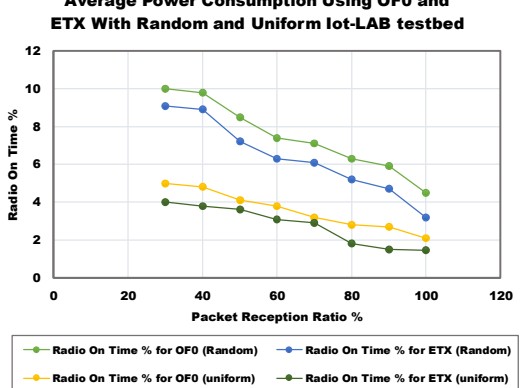

**Figure 14.** The average power consumption in terms of radio on time for OF0 and ETX with uniform and.

*6.2. Cooja Simulated Scenario*

To provide RPL performance in large density networks, we proposed a random scenario of 1000 nodes using Cooja simulator to measure the RPL performance while using OF0 and ETX. The results of Cooja simulations are listed as follows.

■  Control Traffic Overhead

In Figure 15, total control traffic is represented under OF0 and ETX while running a random simulation of the RPL process during 1300 s simulation time. Routes calculated using OF0 need more control messages than ETX as in dense networks. The number of hops is increased while the network size increased. Control messages are used to update the routing table in case of OF0; routing entries to the destination is increased with the network size.

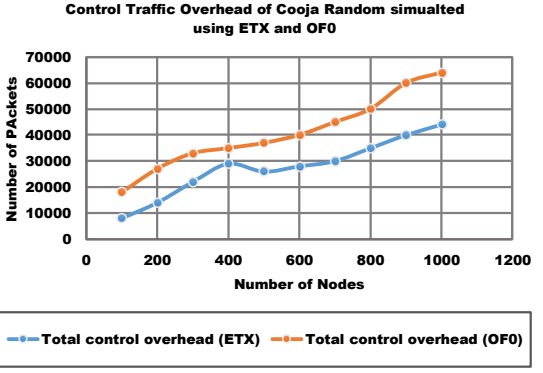

**Figure 15.** The total control traffic in Cooja simulator for OF0 and ETX with random RPL topology.

■  Packet Delivery Ratio (PDR)

The higher the value of PDR, the lower the value of retransmission which leads to less resource waste. Figure 16 shows the variation of PDR value with respect to the network size using OF0 and ETX. It shows that as the network size increases. The PDR decrease and ETX outperform OF0 in route calculation. Hop count OF delivers from 65–35% as the number of nodes increases to reach 1000 while ETX delivers 80–55%.

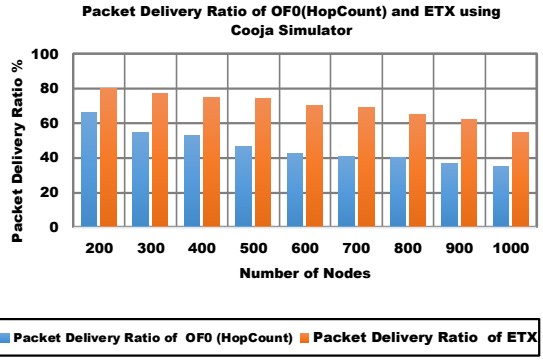

**Figure 16.** packet delivery in Cooja simulator using OF0 and ETX with random network topology.

■  Network Latency (End-to-End delay)

Figure 17 presents network latency while packets travel from client's nodes to the root node (MP2P) traffic using OF0 and ETX in Cooja simulated scenario. In this paper, we consider the time delay of the successful packet delivery, packets retransmission, and packets buffering. However, the network becomes denser and the number of hops increased, and still the rank values using OF0 provide low latency than ETX. As the number of hops is still fixed, OF0 can find the best path with minimum hops quickly rather than using ETX, which based on more retransmissions as the network size increased. ETX and OF0 nearly have the same behavior as the network size, still small, while OF0 outperforms ETX in large-scale networks.

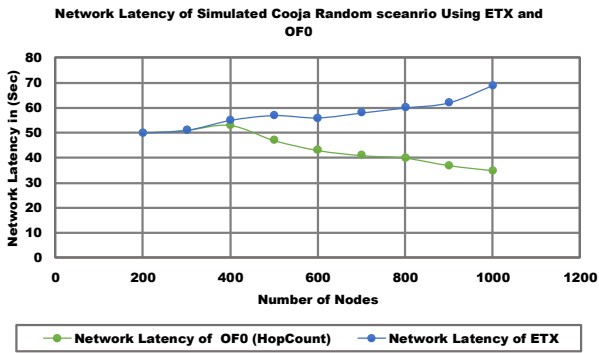

**Figure 17.** Latency comparison using OF0 and ETX with random Cooja simulator.

■  Energy Consumption

Figure 18 illustrates the radio on-time percentage using OF0 and ETX with Cooja random scenario. The radio on-time percentage is increased as the network density increased, which reflect the power consumption during the network for all client nodes. ETX has small radio on-time percentage than OF0, as choosing the best path using hop count in large density network consume a lot of energy to calculate the paths, update the routing tables and share the routing information and candidate parents.

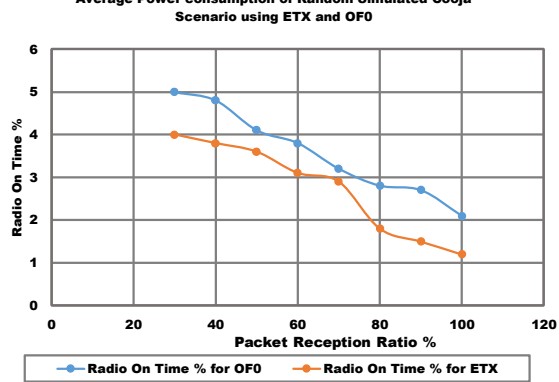

**Figure 18.** Average power consumption in Cooja simulator for OF0 and ETX with random RPL topology.

### 6.3. Results Discussion

All results are summarized and listed in Tables 4–6.

Table 4 shows IoT-LAB results for random and uniform 140 M3 wireless nodes while adopting two different objective functions (OF0 and ETX). Experiments in IoT-LAB lasted for 2 h with sending rate of 45 s and with 70 runs. The measured control traffic overhead is considered as the total overhead traffic for all the network nodes during the experiment time. ETX Objective Function is outperforming OF0 in the defined measured metrics as it has the lowest control overhead, latency, and power consumption while providing higher packet delivery ratio than OF0 overall RPL performance, while the network is uniformly distributed and outperforms the results of random network topology. Random networks experienced high latency, more retransmission, network congestion, and higher power consumption.

**Table 4.** RPL routing metrics results in IoT-lab testbed measured over simulation time.

| Routing Metric (Total nods=140) | Uniform Topology | | Random Topology | |
| --- | --- | --- | --- | --- |
| OF | OF0 | ETX | OF0 | ETX |
| Control Traffic Overhead (PKTs) | 78000 | 62000 | 90000 | 70000 |
| PDR % | 70% | 85% | 45% | 55% |
| Network Latency (S) | 3.5 | 3 | 7 | 5 |
| Average Energy Consumption (mW) | 2.5 | 1.5 | 4.5 | 3.5 |

Table 5 shows simulated results of Cooja simulator for a large-scale network using a random network topology while applying OF0 and ETX. Simulation lasted for 1300 s with sending rate 30 s for 1000 nodes, using only one sink node with one RPL DODAG instance. The Cooja simulation output indicates many variations of RPL performance as the network size increased. It reflects higher latency when RPL protocol uses ETX to calculate the routes and less power consumption with the higher delivery ratio. If RPL proposed a high-density network application, which requires low latency like medical monitoring applications, it is recommended to apply OF0, which basically depend on the number of hops to calculate the routing routes as it experiences lower latency than ETX.

**Table 5.** RPL routing metrics results within cooja simulator measured over the simulation time.

| Routing Metric (Nods No=1000) | Cooja Simulated Topology | |
| --- | --- | --- |
| | OF0 | ETX |
| Control Traffic Overhead (PKTs) | 70,000 | 43,000 |
| (PDR) % | 45% | 66% |
| Network Latency (S) | 38 | 70 |
| Radio on Time % | 2.2% | 1.1% |

**Table 6.** Summary of the comparison of different RPL evaluation environments.

| RPL implementations | | Accettura et al. | Heurtefeux et al. | Hakeem et al. | Baccelli et al | Our Proposed Evaluation |
|---|---|---|---|---|---|---|
| RPL environment | WSN | ✓ | ✓ | ✓ | ✓ | ✓ |
| | PLC | × | × | × | × | × |
| | Simulation | ✓ | × | × | × | ✓ |
| | Experimental | × | ✓ | ✓ | ✓ | ✓ |
| Network scale | Simulation | 100 | × | × | × | 1000 |
| | Experimental | × | 100 | 100 | 27 | 140 |
| RPL OF metric | ETX | ✓ | ✓ | × | × | ✓ |
| | Hop Count | × | × | ✓ | ✓ | ✓ |
| The measured RPL performance metrics | Latency | ✓ | × | × | × | ✓ |
| | Overhead | ✓ | ✓ | ✓ | ✓ | ✓ |
| | PDR | × | × | ✓ | × | ✓ |
| | Power consumption | × | × | × | × | ✓ |
| Network Topology | Uniform | ✓ | ✓ | ✓ | ✓ | ✓ |
| | Random | × | × | × | × | ✓ |

To be fair, we choose from literature some related work [7,9,12,17] that evaluate RPL protocol using operating system Contiki like our study and based WSN environment in terms of routing metrics and targeting many different platforms and scenarios to test RPL applicability in AMI applications. We compare our RPL evaluations using the experimental network and simulated network with these related works and find that we measured RPL performance using two important objective functions (OF0 and ETX) in terms of four routing metrics (overhead, latency, PDR and power consumption). Accettura et al. [7], find that RPL overhead traffic is very high and should be decreased. They didn't measure any other metrics and they use only ETX to find the best path. Their evaluation does not reflect anything about RPL behavior in dense networks or the effectiveness of network physical topology. Heurtefeux et al. [9] evaluate RPL in uniform topology and their results showed that RPL overhead is very high, while they mentioned that RPL performance is not affected by the physical topology. In Hakeem et al. [17] this implementation used hop count to find the best path in medium scale network. Their results lack the RPL end-to-end delay calculation and don't provide RPL behavior in random WSN topology. In Baccelli et al. [12] they provide small network of 27 nodes and they measure only the overhead. In our proposed evaluation, we study RPL routing protocol using hop count and ETX in terms of overhead traffic, end-to-end delay, PDR and power consumption through two different network topologies. From our experimental and simulated scenarios, we found the following:

1. Compared to the mentioned related work [7,9,12,17], we tested the performance using several scenarios and network parameters like network density, routing metrics, physical topologies.
2. In contrary to [7], we figured out that RPL performance strongly depends on the physical network topology. For instance, in the random topology case, we experienced performance degradation in terms of traffic overhead, latency, power consumption and PDR compared to the uniform topology.
3. ETX is outperforming hop count as it introduces little traffic overhead, latency, and power consumption while increasing the PDR value than hop count.
4. For large scale networks, ETX introduces higher latency than Hop Count, thus ETX is only not applicable for low latency applications. Table 6 summarizes the comparison of different RPL evaluation environments.

## 7. Conclusion and Future Work

Most of the SG devices are nowadays microcontrollers with limited storage and computing capabilities. Communication frontend is moreover realized by low bandwidth and low power technologies. Therefore, the design of communication network solutions which satisfy the requirements of these lossy and constrained devices is considered as a great challenge. In this paper,

we have explored the relevance and changes of RPL protocol during a typical AMI preparation. Experiments are conducted using two different hardware platforms with a uniform and random network topology. RPL is tested under different configuration parameters and different network environments using practical and simulated implementation.

RPL performance is depending on the network topology and the number of nodes. This is contrasted with the results of this article [7] as their results introduced that RPL doesn't depend on the distribution of nodes. AMI networks as part of SG communication infrastructure need reliability and low latency communication. RPL protocol satisfies application necessities through an appropriate Objective Function definition.

In this paper, we investigated the control overhead, latency, packet delivery ratio, and power consumption while optimizing two OFs (OF0 and ETX); we intend to outline a simple hop count based version of OF0 to satisfy AMI network necessities and compare it with ETX. Practical implementation using (uniform & random) IoT-LAB topology outlines that routes using ETX outperform hop count as the network size affects the transmissions and calculated the best path using hop count consume more network resources. To measure the RPL performance within a large scale network, a random simulated Cooja scenario is implemented to evaluate RPL performance in terms of some defined metrics while applying ETX and OF0. Results show that the proposed ETX outperform OF0 in packet delivery ratio and power consumption. OF0 experience higher latency than using ETX: As the network size increase, the number of hop count increases.

In the future, we intend to study the RPL performance in PLC environments under a new configuration and modification of the RPL basic parameters to participate in the standardization process of RPL as IPv6 in SG applications.

**Author Contributions:** S.A.A.H. did the experiments, data collection, the real testing, conceptualization, software implementation and writing of the original draft. A.A.H. conceptualization, paper editing and review. H.K. project administrator, editing, reviewing and funding.

**Funding:** This work was supported in part by IITP Grant through the Korean Government, under the development of wide area driving environment awareness and cooperative driving technology which are based on V2X wireless communication under Grant R7117-19- 0164, and in part by the Center for Integrated Smart Sensors funded by the Ministry of Science, ICT & Future Planning as Global Frontier Project, South Korea, (CISS-2018).

**Conflicts of Interest:** The authors declare no conflict of interest.

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
