# Peer review of "RPL Routing Protocol Performance in Smart Grid Applications Based Wireless Sensors: Experimental and Simulated Analysis"

_electronics, doi:10.3390/electronics8020186_

Round 1

Reviewer 1 Report

The paper addressed an interesting topic, but  there are some serious drawbacks that authors should remove:

1) Authors have forgotten the smartmeter technology that have been deployed along the world, using PLC communications and PRIME protocol. Authors should compare RPL performance with the real performance of millions of nodes working with PRIME and similar routing protocols.

2) Some affirmations that appear in the paper are  not correct: for example, smartgrid is not a new energy technology. Smartgrid is the application of communication and information technology to energy grids. Similary, the millions of smart meters allready deployed along de world do not met some of the  highligthed specifications in the paper.

3) There are numerous typos and the quality of most figures is poor.

4) The results are not coherent, as in the IoT-LAB the latency values are above 3000 s using only 140 nodes, while in the cooja simulation the latency is reduced to 38 s using 1000 nodes. What are the reasons?. Moreover a latency of 3000s is not acceptable for real scenarios.

Author Response

The reviewer comments were highly insightful, and they enabled us to improve the quality of our manuscript. In the following pages, we provide our responses to each comment of the reviewers, the changes in manuscript can be tracked as we use the editing mode. According to reviewers, we enhanced the introduction and literature by adding in-depth related work to clarify many points. results and conclusions are enhanced  and a comparison with the previous related work is done. kindly be informed that we changed the title of the paper to clearly expressed its content, we must write that our paper scope targeting AMI applications based WSN not PLC communication. We hope that this revision of the manuscript and the accompanying response notes satisfy the reviewers’ requirements and make our manuscript suitable for publication

Sincerely,

Shimaa kotb, Anar Abdelhady and HyungWon Kim

Reviewer 2 Report

The authors present a very interesting work. They present a quality papaer, well organized and of great interest to readers.
This reviewer considers that the work should be published, but after some minor change:

- In the introduction, on line 44, when talking about PLC, I suggest adding the following reference:

Study of Unwanted Emissions in the CENELEC-A Band Generated by Distributed Energy Resources and Their Influence over Narrow Band Power Line Communications. Energies 2016, 9(12), 1007; https://doi.org/10.3390/en9121007

Author Response

(The authors gave the same response as above.)

Round 2

Reviewer 1 Report

The paper has been considerably improved.

There are still some typographical errors and figures 7 to 11 are difficult to read.